A walk in the maze: variation in Late Jurassic tridactyl dinosaur tracks from the Swiss Jura Mountains (NW Switzerland)

Castanera Diego 1 dcastanera@hotmail.es
http://orcid.org/0000-0002-0135-4455 Belvedere Matteo 2
Marty Daniel 2 3
Paratte Géraldine 2
Lapaire-Cattin Marielle 2
Lovis Christel 2
http://orcid.org/0000-0002-5069-8924 Meyer Christian A. 4
1 Bayerische Staatssammlung für Paläontologie und Geologie, GeoBioCenter, Ludwig-Maximilians-Universität , Munich , Germany
2 Section d’archéologie et paléontologie, Paléontologie A16, Office de la culture , Porrentruy , Switzerland
3 Naturhistorisches Museum Basel , Basel , Switzerland
4 Department of Environmental Sciences, University of Basel , Basel , Switzerland
Marsicano Claudia
Electronic publication date: 2018 Apr 2
Publication date: 2018
Volume: 6
Electronic Location ID: e4579
Received 2017 Dec 6; Accepted 2018 Mar 7
Copyright: © 2018 Castanera et al.
Copyright year: 2018
Copyright holder: Castanera et al.
License: This is an open access article distributed under the terms of the Creative Commons Attribution License, which permits unrestricted use, distribution, reproduction and adaptation in any medium and for any purpose provided that it is properly attributed. For attribution, the original author(s), title, publication source (PeerJ) and either DOI or URL of the article must be cited.
License URL: https://creativecommons.org/licenses/by/4.0/

Keywords: Dinosaur ichnology, Theropods, Kimmeridgian, Reuchenette Formation, Late Jurassic, Switzerland

Funding: Swiss Federal Roads Office Canton Jura Alexander von Humboldt Foundation (Europe Research Stay and Humboldt Research Fellowship for Postdoctoral Researchers) Excavations, scientific documentation of Highway A16 dinosaur tracksites and related research by the Paléontologie A16 (Section d’archéologie et paléontologie, Office de la culture) are funded by the Swiss Federal Roads Office (FEDRO, 95%) and the Canton Jura (5%). Diego Castanera has been supported by the Alexander von Humboldt Foundation (Europe Research Stay and Humboldt Research Fellowship for Postdoctoral Researchers). The funders had no role in study design, data collection and analysis, decision to publish, or preparation of the manuscript.

==============================
Background

Minute to medium-sized (footprint length (FL) less than 30 cm) tridactyl dinosaur tracks are the most abundant in the Late Jurassic tracksites of Highway A16 (Reuchenette Formation, Kimmeridgian) in the Jura Mountains (NW Switzerland). During excavations, two morphotypes, one gracile and one robust, were identified in the field. Furthermore, two large-sized theropod ichnospecies (Megalosauripus transjuranicus and Jurabrontes curtedulensis) and an ornithopod-like morphotype (Morphotype II) have recently been described at these sites.

Methods

The quality of morphological preservation (preservation grade), the depth of the footprint, the shape variation, and the footprint proportions (FL/footprint width (FW) ratio and mesaxony) along the trackways have been analyzed using 3D models and false-color depth maps in order to determine the exact number of small to medium-sized morphotypes present in the tracksites.

Results

The study of footprints (n = 93) recovered during the excavations has made it possible to identify and characterize the two morphotypes distinguished in the field. The gracile morphotype is mainly characterized by a high FL/FW ratio, high mesaxony, low divarication angles and clear, sharp claw marks, and phalangeal pads (2-3-4). By contrast, the robust morphotype is characterized by a lower FL/FW ratio, weaker mesaxony, slightly higher divarication angles and clear, sharp claw marks (when preserved), whereas the phalangeal pads are not clearly preserved although they might be present.

Discussion

The analysis does not allow the two morphotypes to be associated within the same morphological continuum. Thus, they cannot be extramorphological variations of similar tracks produced by a single trackmaker. Comparison of the two morphotypes with the larger morphotypes described in the formation (M. transjuranicus, J. curtedulensis, and Morphotype II) and the spatio-temporal relationships of the trackways suggest that the smaller morphotypes cannot reliably be considered as small individuals of any of the larger morphotypes. The morphometric data of some specimens of the robust morphotype (even lower values for the length/width ratio and mesaxony) suggest that more than one ichnotaxon might be represented within the robust morphotype. The features of the gracile morphotype (cf. Kalohipus isp.) are typical of “grallatorid” ichnotaxa with low mesaxony whereas those of the robust morphotype (cf. Therangospodus isp. and Therangospodus? isp.) are reminiscent of Therangospodus pandemicus. This work sheds new light on combining an analysis of variations in footprint morphology through 3D models and false-color depth maps, with the study of possible ontogenetic variations and the identification of small-sized tridactyl ichnotaxa for the description of new dinosaur tracks.

Introduction

Since the first sauropod tracks were reported in the Lommiswil quarry (Late Kimmeridgian, Canton Solothurn) in the Swiss Jura Mountains (Meyer, 1990), dinosaur track discoveries have increased considerably, and to date more than 25 tracksites have been documented in the cantons of Jura, Bern, Neuchâtel, and Solothurn. Most of these tracksites occur in the Reuchenette Formation (Kimmeridgian), and some of them in the Twannbach Formation (Tithonian) (Meyer & Thüring, 2003; Marty, 2008; Marty & Meyer, 2012; Marty et al., 2010, 2013). Between 2002 and 2011, six large tracksites were systematically excavated and documented by the Palaeontology A16 (Marty & Billon-Bruyat, 2009) prior to the construction of Highway A16 in the Canton Jura (NW Switzerland). These tracksites covered together a surface area of 18,500 m2, and a total of 59 ichnoassemblages. Out of 14,000 individual tracks, 254 trackways were attributed to sauropods and 411 to bipedal tridactyl dinosaurs. Therefore, the Jura carbonate platform has today become a key area for Late Jurassic dinosaur palaeoichnology (Marty, 2008; Marty & Meyer, 2012) as it represents one of the areas with the highest number of Late Jurassic dinosaur tracks in the world.

Recent papers have described giant theropod tracks (Jurabrontes curtedulensis, Marty et al., 2017) and large theropod tracks (Megalosauripus transjuranicus, Razzolini et al., 2017) from the Swiss Jura Mountains, but most of the tridactyl tracks by far are the still largely undescribed minute, small and medium-sized tracks (footprint length (FL) < 30 cm). Marty (2008) described minute and small tridactyl tracks from the Chevenez–Combe Ronde tracksite (Canton Jura, NW Switzerland) and tentatively attributed some of these to Carmelopodus. Since then, however, many other tracksites and ichnoassemblages with minute to medium-sized tridactyl tracks have been discovered, including some very well-preserved tracks of different morphotypes and some very long trackways (up to 100 m).

In Europe, apart from the Swiss and French (Mazin, Hantzpergue & Pouech, 2016) Jura Mountains, the main Late Jurassic deposits that have yielded minute to medium-sized tridactyl dinosaur tracks are located in the Lusitanian Basin in Portugal (Antunes & Mateus, 2003; Santos, 2008), the Asturian Basin in Spain (Lockley et al., 2008; Piñuela, 2015), the Aquitanian Basin in France (Lange-Badré et al., 1996; Mazin et al., 1997; Moreau et al., 2017), the Lower Saxony Basin in NW Germany (Kaever & de Lapparent, 1974; Diedrich, 2011; Lallensack et al., 2015), and several units in the Holy Cross Mountains in Poland (Gierliński, Niedźwiedzki & Nowacki, 2009). The units that date to around the Jurassic–Cretaceous boundary (Tithonian–Berriasian) in the Iberian range in Spain (Santisteban et al., 2003; Castanera et al., 2013a; Alcalá et al., 2014; Campos-Soto et al., 2017) should also be mentioned. It is noteworthy that, while there is a high number of small to medium-sized tridactyl tracks (assigned to both theropods and ornithopods) described, only few ichnotaxa have been formally erected so far. Besides the tracks from the Chevenez–Combe Ronde tracksite tentatively assigned to Carmelopodus by Marty (2008), the main small to medium-sized tridactyl tracks identified have been from Spain (Grallator and Anomoepus, from several sites in Asturias, Lockley et al., 2008; Piñuela, 2015; Castanera, Piñuela & García-Ramos, 2016), France (Carmelopodus, Loulle tracksite, Mazin, Hantzpergue & Pouech, 2016), Poland (Wildeichnus, cf. Jialingpus, and Dineichnus, different units in the Holy Cross Mountains, Gierliński, Niedźwiedzki & Nowacki, 2009), Germany (Grallator, Bergkirchen tracksite, Diedrich, 2011), and Portugal (Dineichnus and ?Therangospodus, Lockley et al., 1998a; Lockley, Meyer & Moratalla, 2000). Other significant Late Jurassic areas with minute to medium-sized tridactyl dinosaur tracks are found in the USA (Foster & Lockley, 2006), Morocco (Belvedere, Mietto & Ishigaki, 2010), China (Xing, Harris & Gierliński, 2011; Xing et al., 2016), Yemen (Schulp & Al-Wosabi, 2012), and Turkmenistan (Lockley, Meyer & Santos, 2000; Fanti et al., 2013).

Several recent papers have examined the variability in track morphology along trackways (Razzolini et al., 2014, 2017; Lallensack, van Heteren & Wings, 2016), showing how pronounced changes can occur along a single trackway. Thus, sometimes it can be very difficult to determine the exact number of ichnotaxa and clearly distinguish between them, especially when the tracks are morphologically similar. This should be borne in mind particularly when studying the material from Switzerland, where large theropod tracks have shown notable variations in shape along the same trackway (Razzolini et al., 2017). In the case of the minute to medium-sized tridactyl tracks, two different morphotypes were identified at first glance during the documentation of the tracksites, one gracile and one more robust type. The aim of this paper is to describe the small to medium-sized tridactyl tracks collected in the Jura Mountains (NW Switzerland). Special emphasis is put on the analysis of track morphology through 3D models and possible variations in footprint shape along trackways in order to find out if the different morphotypes are a consequence of preservation. In addition, other factors such as possible ontogenetic variations in the larger ichnospecies described in the formation are also taken into account. Finally, we discuss the ichnotaxonomy of the tracks.

Geographical and Geological Setting

The studied material comes from six different tracksites from Highway A16 and nearby areas (Fig. 1A): (1) Courtedoux–Bois de Sylleux (CTD–BSY), (2) Courtedoux–Tchâfouè (CTD–TCH), (3) Courtedoux–Béchat Bovais (CTD–BEB), (4) Courtedoux–Sur Combe Ronde (CTD–SCR), (5) Chevenez–Combe Ronde (CHE–CRO), and (6) Chevenez–La Combe (CHE–CHV). For the sake of simplicity BSY, TCH, BEB, SCR, CRO, and CHV are used in the publication.

Figure 1 Geographical and geological settings of the Highway 16 tracksites (modified from Razzolini et al., 2017; Marty et al., 2017).

(A) Geographical setting of the Ajoie district (NW Switzerland) with the location of the tracksites (1, Courtedoux—Béchat Bovais; 2, Courtedoux—Bois de Sylleux; 3, Courtedoux—Tchâfouè; 4, Courtedoux—Sur Combe Ronde; 5, Chevenez—Combe Ronde; 6, Chevenez—La Combe) along Highway A16. (B) Chrono-, bio-, and lithostratigraphic setting of the Reuchenette Formation in the Ajoie district, Canton Jura, NW Switzerland (after Comment, Ayer & Becker, 2011; Comment et al., 2015).

All the tracksites are located in the Ajoie district about 6–8 km to the west of Porrentruy (Canton Jura, NW Switzerland) on the track of Swiss federal highway A16 except the Chevenez–La Combe tracksite, which is located in a quarry near the village of Chevenez. The first five tracksites were systematically excavated level-by-level by the Palaeontology A16 (PALA16) from 2002 to 2011 (Marty et al., 2003, 2004, 2007; Marty, 2008; Marty & Billon-Bruyat, 2009).

Geologically, the study area belongs to the Tabular Jura Mountains and is located at the eastern end of the Rhine–Bresse transfer zone between the Folded Jura Mountains (South and East) and the Upper Rhine Graben and Vosges Mountains (North). The Upper Jurassic strata of the Swiss Jura Mountains are made up of shallow-marine carbonates deposited on the large and structurally complex Jura carbonate platform, which was located at the northern margin of the Tethys at a palaeolatitude of approximately 30° N (Thierry, 2000; Thierry et al., 2000; Stampfli & Borel, 2002).

The tracksites occur within the Reuchenette Formation (Kimmeridgian), dated by ammonites of the Cymodoce to Mutabilis (Boreal), and Divisum to Acanthicum (Tethyan) biozones (Comment et al., 2015). Accordingly, the age of the track-bearing levels is late Early to early Late Kimmeridgian (Gygi, 2000; Jank, Wetzel & Meyer, 2006). This age is also confirmed by the presence of ostracods (Schudack et al., 2013). More information on the sedimentology and palaeoenvironment of the Highway A16 tracksites can be found in Jank, Wetzel & Meyer (2006), Marty (2008), Razzolini et al. (2017), and Marty et al. (2017).

Stratigraphically, the tracksites include three different track-bearing laminite intervals, separated by shallow marine limestones (Marty, 2008; Waite et al., 2008; 2013; Comment, Ayer & Becker, 2011; Comment et al., 2015). The three main track-bearing laminite intervals are referred to as the lower, intermediate and upper levels, respectively, levels 500–550, 1,000–1,100, and 1,500–1,650 (Fig. 1B). Only tracks from the lower and intermediate track levels are included in the present study (Fig. 1B), and the studied tracks come from a total of 11 different ichnoassemblages (stratigraphic track levels). These are as follows: BEB500, CRO500, BSY1020, BSY1040, BSY1050, TCH1055, SCR1055, TCH1060, TCH1065, TCH1069, and CHV1000–1100 (precise level cannot be indicated).

Material and Methods

We analyzed a total of 93 individual tracks (Table S1) that are housed in the track collection of PALA16 (Canton Jura), either as original specimens or as replicas. This collection will be transferred to JURASSICA Muséum (Porrentruy, Canton Jura) in 2019. All the tracks are from the aforementioned tracksites, the largest samples coming from BEB500 (39 footprints), TCH1065 (15), and CRO500 (20). Each analyzed track has two acronyms (Table S1): one represents the number of the slab within the collection, e.g., TCH006-1100 denotes Tchâfouè tracksite, year 2006 (the year of discovery), slab 1,100 (an “r” in front of the specimen number, means replica). Some high-resolution laser scans were made in the field and those tracks are here referred to as “Laser-Scan”. A second acronym represents the level and number of the trackway and track, e.g., TCH1055-T2-L1 denotes Tchâfouè tracksite, level 1,055, trackway 2, track 1, left pes. The second acronym is used throughout the manuscript. As the track-bearing layers were excavated level-by-level there are no doubts about the preservation mode of the tracks. Thus, all the tracks were preserved as true tracks (concave epireliefs) and were produced on the tracking surface, with the only exception of TCH1060-E58, which was preserved as a natural cast (convex hyporelief).

Analysis of track morphology was performed independently for each track; however, some tracks belong to trackways, therefore their variation in morphology along a single trackway was also considered in order to avoid over-identification of morphotypes. These trackways are: BEB500-T16 (3), BEB500-T17 (4), BEB500-T58 (6), BEB500-T73 (4), BEB500-T75 (2), BEB500-T78 (2), BEB-500-T82 (2), BEB-500-T93 (2), BEB500-T120 (4), CRO500-T10 (14), CRO500-T30BIS (5), TCH1055-T2 (2), TCH1065-T15 (2), TCH1065-T25 (2), and TCH1069-T2 (2). We analyzed each individual track and made an evaluation of the quality of preservation according to the scale of Belvedere & Farlow (2016) (Table S1). As stated by these authors, “quantitative shape analyses need to be based on data of high quality, and comparisons are best made between tracks comparable in quality of preservation.” Accordingly, only the tracks with a preservation grade equal to or higher than two were considered for measurement and analyzed in this paper; field measurements exist for all the other tracks and are stored in the PALA16 database. The descriptions are based on identification of two different morphotypes, one gracile and one robust, during the documentation in the field. Thus, the FL, footprint width (FW), length and width of digits II (LII, WII), III (LIII, WIII), and IV (LIV, WIV), divarication angles (II–III; III–IV) were measured (see Castanera, Piñuela & García-Ramos, 2016, fig. 2). Subsequently, the FL/FW ratio and the mesaxony were calculated. The latter was calculated on the basis of the anterior triangle (AT) length–width ratio following Lockley (2009). All these measurements were taken from perpendicular pictures with the software Image J. The tracks were classified according to different size classes (Marty, 2008) on the basis of pes length (FL) as: (1) minute, FL < 10 cm; (2) small, 10 cm < FL < 20 cm; (3) medium, 20 cm < FL < 30 cm; and (4) large, FL > 30 cm. The morphometric data of the studied tracks were compared in a bivariate plot (length/width ratio vs. mesaxony) with larger tracks (M. transjuranicus, J. curtedulensis, and Morphotype II) described in the Reuchenette Formation (Razzolini et al., 2017; Marty et al., 2017). In addition, they were also compared with other theropod ichnotaxa using data from Castanera, Piñuela & García-Ramos (2016) which were mainly compiled after Lockley (2009) and Xing et al. (2014). Data were analyzed with the software PAST v.2.14 (Hammer, Harper & Ryan, 2001).

3D-photogrammetric models were generated from pictures taken with a Canon EOS 70D camera equipped with a Canon 10–18 mm STL lens using Agisoft Photoscan (v. 1.3.2, www.agisoft.com) following the procedures of Mallison & Wings (2014) and Matthews, Noble & Breithaupt (2016). Within the BEB500 sample, 3D data of 10 footprints were obtained by high-resolution laser-scanning carried out in the field in 2011 by Pöyry AG with a Faro hand-scanner. Most of these 10 footprints were destroyed during the construction of Highway A16. The scaled meshes were exported as Stanford PLY files (.ply) and then processed in CloudCompare (v.2.7.0, www.cloudcompare.com) in order to obtain accurate false-color depth maps. All photogrammetric meshes used in this study are available for download here: https://doi.org/10.6084/m9.figshare.5662306.v2 (ca. 2.5 Gb). In addition, we analyzed the maximum depth of all the tracks, in order to ascertain whether there is a relationship between depth, preservation grade, and the morphotype. The maximum depth was estimated using the false-color map derived from the 3D-model in those tracks with a preservation grade generally higher than 0.5.

Description of the Track Morphotypes and Morphological Variations along the Trackways

Gracile morphotype

This morphotype was identified in all six tracksites. The footprints are small to medium-sized (15–21.2 cm) tridactyl tracks (Fig. 2), clearly longer than wide (FL/FW ratio = 1.50–1.90) (Table 1). The digits are slender with an acuminate end and clear claw marks preserved in the three digits in the majority of the tracks. Digit III is clearly longer and slightly wider than digits II and IV. Digits II and IV are similar in length and width. The mesaxony is variable but medium to high (AT = 0.53–0.98), with a mean value of 0.77, although it is higher in most of the specimens (more than 0.8 in half of the sample). The divarication angles are relatively low, II–III generally being slightly higher (mean 25°) than III–IV (mean 22°). The hypices are quite symmetrical. The “heel” morphology is variable; some specimens have an oval to round heel pad connected with digit IV (BEB500-T16-R3, TCH1055-E53, TCH1055-T2-R1, TCH1069-T1-R2; see Fig. 2), whereas in others it is not clearly preserved even when the preservation grade is high (e.g., BSY1020-E2). Most of the specimens preserve a clear small medial notch located behind digit II, which with the rounded heel marks gives them an asymmetric shape. In some of the footprints well-defined digital pads can be discerned. The tracks with the best quality of preservation suggest a phalangeal formula of 2-3-4 (including the metatarsophalangeal pad IV).

Figure 2 Pictures and false-color depth maps of the tracks with a high preservation grade that belong to the gracile morphotype.

(A) BEB500-T16-R3; (B) BEB500-T26-R5; (C) BEB500-T73-L5; (D) BSY1020-E2; (E) CHV1000-E4; (F) CRO500-T10-L10; (G) SCR1055-T2-L2*; (H) SCR1055-T3-L2*; (I) TCH1055-E53; (J) TCH1055-T2-L1; (K) TCH1060-E58; (L) TCH1065-E3; (M) TCH1065-E177; (N) TCH1065-T25-L2; (O) TCH1069-T1-R2. *In these two cases, it is not a picture but a colored mesh obtained from the 3D-model. Scale bar = 5 cm.

Table 1 Measurements of the specimens with a high preservation grade.

Track	FL	FW	FL/FW	LII	LIII	LIV	WII	WIII	WIV	II–III	III–IV	ATw	Atl	AT	
BEB500-T16-R3	18	10	1.8	13.5	18	13.8	2	1.9	1.8	22.5	17.5	8.8	5.8	0.66	
BEB500-T17-R8	19	11.5	1.65	11	19	13	1.9	3.3	1.6	23	20	10.5	8.8	0.84	
BEB500-T26-R5	19	12	1.58	13	19	14	2.2	3	2.9	32	26	10.3	9.4	0.91	
BEB500-T73-L5	15	8.5	1.76	8.5	15	10	2.3	2.9	2.5	31	22	7.9	5.8	0.73	
BSY1020-E2	22	11.7	1.88	15	22	13.5	3.6	3	2.7	21.5	24.5	9.5	8.5	0.89	
TCH1055-E53	17.5	10.3	1.7	12.2	17.5	12	3	2.7	2.5	25	17.5	8.5	7	0.82	
TCH1055-T2-L1	21.2	13.1	1.62	15.6	21.2	15	2.3	2.1	2.2	25	22	11.4	7	0.61	
TCH1055-T2-R1	19.5	13	1.5	13.2	20.5	13.1	3.3	3.7	2.5	29	23	10.6	8.5	0.80	
TCH1060-E58	20	10.5	1.90	20	13.5	12	3.4	3.1	2.9	27	22	8.8	7.5	0.85	
TCH1065-E177	17.5	9.4	1.86	11.8	17.5	12.5	1.6	2.4	2	21	20	8.2	6.5	0.79	
TCH1065-E3	18.4	12.3	1.5	12.3	18.4	11.7	3.3	3.8	2.3	30	24	9.14	7.8	0.85	
TCH1065-T25-L2	19.3	12.2	1.58	14	19.3	12.3	3	3	2.7	25	21	10.3	8	0.78	
TCH1069-T1-R2	20	13	1.54	14	20	13.5	2.1	2.7	2.1	24	29	11.5	8.3	0.72	
SCR1055-T2-L2	20	12	1.67	15	20	16	2.7	2.9	2.5	25	18	11.4	6	0.53	
SCR1055-T3-L2	18	11	1.64	12	18	12	2.3	2.1	1.8	26	26	8.5	8.3	0.98	
CHV1000-E4	16	8.5	1.88	11	16	10	1.8	2.3	1.7	21	22	8.1	6.1	0.75	
CRO500-T10-L10	11	6.5	1.69	6	11	7	1.4	1.8	1.5	32	23	5.6	5.4	0.96	
BEB500-T120-R5	17	15	1.13	13.5	17	14.5	3.5	3.2	2.5	30.4	34	13	5	0.38	
BEB500-T120-R6	18	15.5	1.16	14.5	18	15	2.5	3.1	3	22	27	14.2	5.7	0.40	
TCH1065-E124	19	15.5	1.23	13.5	19	15	3.3	4.5	3.5	27.5	26.5	14.4	7.5	0.52	
TCH1065-E188	18	12.3	1.46	13.3	18	13	3.2	3.7	3.3	25	27	10	5.2	0.52	
TCH1065-T21-R1	19.8	14.5	1.37	14.4	19.8	14.8	3.5	3.7	3.5	27	27	11.8	6.9	0.58	
TCH1065-T15-R1	21.8	15	1.45	15.7	21.8	17.2	2.7	3.4	3.1	29	25	12	7.3	0.61	
Notes:

(FL), footprint length; (FW), footprint width; (FL/FW), footprint length/footprint width ratio; (LI, LII, LIII), digit length; (WI, WII, WIII), digit width; (II–III, III–IV), divarication angles; mesaxony (AT, anterior triangle ratio; ATw, anterior triangle width; ATl, anterior triangle length).

Robust morphotype

This morphotype has mainly been identified on the track levels BEB500 and TCH1065. The footprints are small or medium-sized (17–21.8 cm) tridactyl tracks (Fig. 3), slightly longer than wide (FL/FW ratio = 1.13–1.46) (Table 1). The digits are relatively robust with an acuminate end and clear claw marks preserved in some of the tracks (e.g., BEB500-T120-R5, TCH1065-T15-R1, TCH1065-T21-R1). Digit III is clearly longer and slightly wider than digits II and IV. Digits II and IV are similar in length and width. The mesaxony is variable but low-medium (AT = 0.38–0.61), with a mean value of 0.49. The divarication angles are low, II–III (mean 26°) and III–IV (mean 27°) being quite similar. The hypices are quite symmetrical. The “heel” morphology is variable, ranging from subrounded to subtriangular. Only TCH1065-T21-R1 preserves a clear small medial notch located behind digit II, thus being slightly asymmetrical, whereas the other specimens are more symmetrical. Well-defined digital pads cannot be discerned in most of the footprints, although TCH1065-T21-R1 shows digital pads suggesting a possible phalangeal pad formula of 2-3-4.

Figure 3 Pictures and false-color depth maps of the tracks with a high preservation grade that belong to the robust morphotype.

(A) BEB500-T120-R5; (B) BEB500-T120-R6; (C) TCH1065-T21-R1; (D) TCH1065-E188; (E) TCH1065-E124; (F) TCH1065-T15-R1. Scale bar = 5 cm.

Table 2 and Figs. 4 and 5 show the variations in preservation grade and the maximum depth along the analyzed trackways (see Table S1 to see the data of each specific track). 11 trackways belong to the gracile morphotype and four trackways belong to the robust morphotype (see Figs. S1–S3 for the location of the trackways in the tracksite). The preservation grade varies from low (0–0.5 in the scale) to high (2 or more in the scale) for both morphotypes. The maximum depth variation is considerably low, as the maximum variation it is around 5 mm in both morphotypes (CRO500-T30BIS and BEB500-T120).

Table 2 Variation in preservation grade and variation in maximum depth (difference between highest and lowest value) through the analyzed trackways (see Table S1 for specific values of each track).

Gracile morphotype	Robust morphotype	
Trackway	Number of tracks	Analyzed tracks	Variation in preservation grade (min–max)	Total variation in depth (mm)	Trackway	Number of tracks	Analyzed tracks	Variation in preservation grade (min–max)	Total variation in depth (mm)	
BEB500-T16	27	3	0.5–2.5 (High)	1.1	BEB500-T75	71	2	0	0	
BEB500-T17	120	4	1–2 (Medium)	2.8	BEB500-T120	29	4	0–2 (High)	5.8	
BEB500-T58	53	6	0.5–1.5 (Medium)	2.3	TCH1065-T15	2	2	0.5–2 (High)	1.5	
BEB500-T73	18	4	1–2 (Medium)	2	TCH1069-T2	5	2	1–1.5 (Low)	1.8	
BEB500-T78	24	2	1–1 (Low)	0.4						
BEB500-T82	59	2	1.5–1.5 (Low)	1.9						
BEB500-T93	64	2	1–1.5 (Low)	3.8						
CRO500-T10	75	14	0–2 (High)	2.6						
CRO500-T30BIS	11	5	0–2 (High)	4.7						
TCH1055-T2	4	2	2–2.5 (Low)	2.5						
TCH1065-T25	4	2	1–2 (Medium)	2.7						

Figure 4 Morphological variation in the footprint shape along the studied trackways from BEB500 tracksite.

(A) BEB500-T16 (gracile morphotype); (B) BEB500-T17 (gracile morphotype); (C) BEB500-T58 (gracile morphotype); (D) BEB500-T73 (gracile morphotype); (E) BEB500-T75 (gracile morphotype); (F) BEB500-T78 (gracile morphotype); (G) BEB500-T82 (gracile morphotype); (H) BEB500-T120 (robust morphotype); (I) BEB500-T93 (gracile morphotype).

Figure 5 Morphological variation in the footprint shape along the studied trackways from the CRO500, TCH1055, TCH1065, and TCH1069 tracksites.

(A) CRO500-T10 (gracile morphotype); (B) CRO500-T30BIS (gracile morphotype); (C) TCH1055-T2 (gracile morphotype); (D) TCH1065-T15 (robust morphotype); (E) TCH1069-T2 (robust morphotype); (F) TCH1065-T25 (gracile morphotype).

Discussion

True ichnodiversity or variation due to substrate-foot interaction?

The final shape of a footprint is determined by a combination of factors related to the anatomy of the trackmaker’s autopodium, the kinematics and the substrate (Marty, Strasser & Meyer, 2009; Falkingham, 2014); another important factor is the level in which the tracks were preserved (Milàn & Bromley, 2006), i.e., if they are preserved as undertracks. In the case of the tracksites of Highway A16, we can rule out this factor as the excavations were carried out level-by-level, so the footprints are true tracks (or natural casts). As the foot-substrate interaction is a major determinant of the final shape of a track, it is important to analyze variations in depth and shape along trackways to ascertain the morphological variation (e.g., Razzolini et al., 2014). For this reason, we first analyzed the individual footprint shape (Figs. 2 and 3) and then looked at the variation along the trackway (Figs. 4 and 5). The idea was to establish whether some of the described morphotypes might represent variations produced by the same/similar trackmakers (in the sense of a theropod with tridactyl functionally similar pes structure, as it is not possible to assign the tracks to a particular clade) walking on a substrate with different properties (water content, thickness or cohesiveness). Previous researchers have described variations in dinosaur footprint shape between two extremes of a morphological continuum or a gradational series (Gatesy et al., 1999; Razzolini et al., 2014) to suggest that similar theropods traversed substrates of variable consistency. Other researchers have shown variations in dinosaur footprint morphology as a consequence of locomotor adaptations associated with changes in substrate consistency (Wilson, Marsicano & Smith, 2009). Thus, the same trackmaker can produce footprints with significant shape variation along the trackways when there is a change in the aforementioned substrate properties. Only in such cases where the analysis is along the trackway, the differences can be claimed as a consequence of foot-substrate interactions rather than anatomical differences in the foot morphology of the trackmaker. In the Swiss samples, clear evidence of intermediate morphologies is missing, supporting the presence of at least two different groups of tridactyl trackmakers. Where gradational series of theropod tracks have been reported (see refs above), these show a hallux, metatarsal marks, and distinctive displacement rims in the deepest tracks that are clearly extramorphological features. None of the morphotypes presented in this paper shows such evidence, even in the deepest tracks. This leads us to think that the sediment was relatively firm during the impression of the tracks.

Generally, tracks with a preservation grade of 1 or higher can be classified in one of the two described morphotypes: gracile or robust. There are just a few classification doubts regarding isolated footprints (e.g., CRO500-T30BIS-R4). At the outset, one possible hypothesis was that the robust morphotype could be a variation of the gracile morphotype, produced by a similar trackmaker on a substrate with different rheological properties (e.g., Gatesy et al., 1999; Razzolini et al., 2014). This hypothesis was especially appealing given the similar footprint dimensions of the two morphotypes. Thus, the deeper tracks would look more robust than the shallow ones, and the absence of clear phalangeal pad marks in most of the robust morphotype tracks might be a consequence of a softer substrate or of deeper penetration by the trackmaker foot. Our max-depth analysis (see Tables 2 and S1), indeed showed that the robust tracks have high values of maximum depth (e.g., BEB500-T120-R5 = 6.1 mm; BEB500-T120-R6 = 10 mm; BEB500-E1 = 10.5 mm; TCH1065-E124 = 6.9 mm; TCH1065-E188 = 5.9 mm; TCH1065-T15-R1 = 8.3 mm; TCH1065-T21-R1 = 12.1 mm, see Table S1). However, it is worth noticing that the highest values all occur on level TCH1065, where also the gracile tracks show their deeper values (TCH1065-E28 = 11.7 mm; TCH1065-T25-R2 = 12.9 mm; TCH1065-T25-L2 = 10.2 mm), quite comparable to the robust ones. Therefore, on this track level the presence of the two morphotypes cannot be directly associated with the depth of the footprints, as both the gracile and the robust show similar values of maximum depth. In the case of BEB500 we see a similar scenario, e.g., BEB500-T16 and BEB500-T17 (gracile) have the same depth as BEB500-T120 and BEB500-E1 (robust). In other words, the depth of the tracks is more determined by the level where they were impressed than by being robust or gracile. Anyway, it is noteworthy that the max-depth analysis shows that depth values are relatively low with maximum depths of slightly more than just 1 cm in the deepest tracks.

The analysis of the morphological variation along the trackways shows that the gracile morphotype is quite consistent along the trackways, and no tracks classifiable as robust are found within these trackways. There are only a few cases, e.g., CRO500-T30BIS-R4 (Fig. 5B) and BEB500-T17-L8/ BEB500-T17-L9 (Fig. 4B), which might look more robust than the other tracks in the trackway, but here the features did not properly fit with the description of the robust morphotype. Regarding the robust morphotype, in the analyzed trackways (BEB500-T120, TCH1065-T15, and TCH1069-T2) none of the tracks shows any feature of the gracile morphotype (noteworthy is the low preservation grade and the scarce data for TCH1065-T15 and TCH1069-T2). The maximum variation in depth along the trackways is low, being around 5 mm in both morphotypes (Gracile, CRO500-T30BIS = 4.7 mm; Robust, BEB500-T120 = 5.8 mm). These data suggest that, in our case, there is no clear correlation between the depth of the footprint and the morphotypes (either gracile or robust), and that the intratrackway variation is never significant enough to denote a shift between the morphotypes. Therefore, the present evidence indicates that there are at least (see following discussion) two different trackmakers of small to medium-sized theropods in the tidal flats of the Jura Mountains.

Analysis of the mesaxony (it represents how far the projection of digit III extends with respect to digits II and IV) and the FL/FW ratio supports the presence of at least the two morphotypes (Fig. 6). Some authors have used mesaxony (Weems, 1992; Lockley, 2009) as a good parameter to distinguish between tridactyl tracks. In the studied sample, this parameter is clearly lower in the robust morphotype than in the gracile one. The FL/FW ratio also shows a considerable difference between the morphotypes (likewise lower in the robust morphotype). A closer look at these two parameters within the robust morphotype (Fig. 6B) raises the question whether it represents a single ichnotaxon. The data for the two analyzed tracks from BEB500-T120 (AT = 0.38–0.40; FL/FW = 1.13–1.16) show considerably lower data for the FL/FW ratio and weaker mesaxony than the tracks from TCH1065, (AT = 0.52–0.61; FL/FW = 1.23–1.46) (see also following discussion).

Figure 6 Bivariate graph plotting the footprint length/footprint width ratio against the mesaxony (AT) of the studied tracks (gracile and robust morphotype) with the larger tracks described in the Reuchenette Formation.

(A) Gracile and robust morphotype compared with Megalosauripus tracks (including tracks classified as Megalosauripus transjuranicus, Megalosauripus cf. transjuranicus and Megalosauripus isp.), the Morphotype II tracks and Jurabrontes curtedulensis (after Razzolini et al., 2017; Marty et al., 2017). Note that in many cases the points represent tracks from the same trackway, so variation through the trackway is also represented. (B) The studied tracks compared with just the holotype and paratype specimens of Megalosauripus transjuranicus and Jurabrontes curtedulensis, plus the best-preserved tracks of Morphotype II (BEB500-TR7). Outline drawings not to scale. The specimen in red is CRO500-T10-L10 (previously classified as Carmelopodus and herein consider as part of the gracile morphotype, see discussion).

Morphotype variation due to ontogeny?

Another salient point relating to the number of morphotypes in the analyzed sample is the possibility of variations due to different ontogenetic states. Few works have dealt with the relationship between dinosaur footprints and ontogeny (e.g., Lockley, 1994; Matsukawa, Lockley & Hunt, 1999; Hornung et al., 2016). Ontogenetic variations have been suggested to explain morphological variation in the classical theropod ichnotaxa of the Grallator–Eubrontes plexus (Olsen, 1980; Olsen, Smith & McDonald, 1998; Moreau et al., 2012). Olsen, Smith & McDonald (1998) proposed that the major proportional differences between Grallator, Anchisauripus, and Eubrontes might be derived from the allometric growth of individuals of several related species. In these typical theropod footprints the large tracks (Eubrontes) are wider with weaker mesaxony than the smaller tracks (Grallator), showing a positive correlation between the elongation of the track and a stronger mesaxony (Lockley, 2009). As this author suggested, the assumption of ontogenetic variation is thus based mainly on the hypothesis of a discernible allometric pattern. Although the growth dynamics have been documented in several groups of theropods (Bybee, Lee & Lamm, 2006; Griffin, 2018, and references therein) little is known about how possible ontogenetic variations may have affected variations in theropod feet proportions and thus in footprint shape (Farlow & Lockley, 1993; Farlow et al., 2013). Generally, tracks that are similar in morphology but different in size are considered to belong to the same ichnotaxon (Thulborn, 1990; Lockley, 1994; Matsukawa, Lockley & Hunt, 1999; Clark, Ross & Booth, 2005; Pascual-Arribas & Hernández-Medrano, 2011; Castanera et al., 2015). Demathieu (1990) also explored the use of ratios of length characters to reduce the influence of size when comparing footprints. For instance, Lockley, Mitchell & Odier (2007) assumed that small theropod tracks (Carmelopodus) from the Jurassic of North America represent adults of small species and not juveniles of larger species and suggested that “this inference is consistent with a model of rapid growth rates such as is typical of birds, which would have reduced the number of potential track making juveniles that could habitually make footprints.” By contrast, Pascual-Arribas & Hernández-Medrano (2011) considered minute theropod tracks from the Early Cretaceous of Spain (subsequently assigned to Kalohipus bretunensis by Castanera et al., 2015) to belong to baby theropods because of the morphometric similarities with larger tracks from the same site and formation.

Two possible different ontogenetic stages should be considered in the interpretation of the Ajoie ichnocoenosis. In one case, there are similarities between the gracile morphotype and the previously described Carmelopodus tracks from the Chevenez–Combe Ronde tracksite (CRO500-T8; CRO500-T10; CRO500-T16; CRO500-T21; CRO500-T26; CRO500-T41) so the first hypothesis would be that both are an ontogenetic variation of the same morphotype. According to the original description by Marty (2008), these tracks can be characterized as mesaxonic, slightly asymmetric, tridactyl tracks that are clearly longer than wide. Digit III is always the longest, digit IV being longer than digit II, which is shorter posteriorly. Claw impressions are present in the three digits, and there is a phalangeal pad formula of 2-3-3. There is a low total divarication angle, and divarication angles of the same order between digits II and III, and III and IV. It has a narrow-gauge trackway with small tracks with outward rotation. CRO500-T10-L10 is the track with the highest preservation grade recovered from level CRO500. Regarding the data taken from this footprint, it should be noted that the FL/FW ratio (1.69) falls within the range of the other gracile tracks, while the mesaxony is among the highest in the whole sample (0.96) but still within the range of the gracile morphotype (Fig. 6). The divarication angle is also low (32°–23°). Moreover, reanalysis of the tracks with the use of false-color depth maps (Fig. 2F) allowed the fourth phalangeal pad in digit IV to be distinguished, suggesting a formula of 2-3-4, although this is not preserved in most of the tracks with a lower preservation grade (Fig. 5A). Accordingly, we consider that there are not enough data to interpret CRO500-T10-L10 as a different morphotype of the tracks included in the gracile morphotype and we regard them as part of it (see next section for the new ichnotaxonomic assignation). This fact highlights the importance of analyzing large samples and the variation in shape through the trackways. The differences in size between CRO500-T10-L10 (FL = 11 cm) and the rest of the specimens of the gracile morphotype (FL = 16–21.2 cm) but similar footprint shape, FL/FW ratio and mesaxony might be explained by an isometric growth. However, the absence of more data of smaller size classes of the gracile morphotype, prevent us to test this first hypothesis.

A second hypothesis considers whether the gracile and the robust morphotype might be ontogenetic variations of one of the three previously identified larger ichnotaxa (M. transjuranicus, J. curtedulensis, and the informally named Morphotype II tracks) of the Jura Mountains (Razzolini et al., 2017; Marty et al., 2017). The two formally described ichnospecies represent large and more slender (M. transjuranicus) and giant and more robust (J. curtedulensis) theropod tracks, respectively. In addition, the third large morphotype not assigned to any known ichnotaxon and named Morphotype II is characterized by subsymmetric tracks that are generally slightly longer than wide (sometimes almost as wide as long), blunt digit impressions, with no evidence for discrete phalangeal pad and claw marks (Razzolini et al., 2017). The interpretation of this morphotype is quite complex, as tracks with the aforementioned features have been also documented as a morphological variation in Megalosauripus trackways. However, other trackways show a very consistent morphology throughout very long trackways, and have been considered by the authors a true unnamed ichnotaxon different from Megalosauripus and with a probable ornithopod affinity. This means that tracks with Morphotype II features can represent two different trackmakers, a theropod (Megalosauripus) and an ornithopod (the proper informally named Morphotype II). Long trackways of Morphotype II are found on the same surfaces that many in the studied sample come from, such as BEB500 and CRO500 (Razzolini et al., 2017). Thus, the hypothesis that the gracile and the robust morphotypes might represent juvenile/subadult specimens of the larger tracks described in the tracksites must be explored.

Analyzing footprint proportions, it should be noted that the FL/FW ratio of the gracile morphotype fits within the upper range of the tracks included in the ichnotaxon Megalosauripus (Fig. 6A) from the Reuchenette Formation; considering just the type material of M. transjuranicus, it fits well (Fig. 6B) (Razzolini et al., 2017). However, the mesaxony is substantially higher in the gracile morphotype than in the Megalosauripus tracks. In the case of the robust morphotype, the FL/FW ratio fits within the range of J. curtedulensis and the Morphotype II tracks when analyzing all the referred material (Fig. 6A) or just the type material of J. curtedulensis and the best-preserved tracks of Morphotype II (BEB500-TR7-L2; BEB500-TR7-R2; BEB500-TR7-R7; BEB500-TR7-L10, Razzolini et al., 2017) (Fig. 6B). The robust morphotype has higher mesaxony than J. curtedulensis, being more similar in this respect to the Morphotype II tracks. It is notable that the footprint proportions within the robust morphotype are quite variable between stratigraphic levels. For example, tracks from trackway BEB500-T120 have a lower FL/FW ratio and mesaxony, whereas tracks from track level TCH1065 have higher ratios. Thus, BEB500-T120 is closer to the ranges of J. curtedulensis whereas the tracks from TCH1065 are closer to the ranges of M. transjuranicus and especially the Morphotype II tracks (Fig. 6).

As previously discussed, the variations in mesaxony, where larger tracks have lower mesaxony, are well documented in theropod tracks (Weems, 1992; Olsen, Smith & McDonald, 1998; Lockley, 2009). Because there are some overlapping areas in the footprint proportions of the larger and the smaller tracks, it might be tempting to relate them according to these values; i.e., gracile with M. transjuranicus, robust from BEB500 with Jurabrontes, and robust from TCH1065 with Morphotype II. Nonetheless, the smaller morphotypes show other considerable morphological differences apart from size and mesaxony with respect to the larger morphotypes. The gracile morphotype differs from M. transjuranicus in key features of the diagnosis such as the sigmoidal impression of digit III (less sigmoidal), the divarication angle (less divaricated), and the digital pad of digit IV (proportionally smaller when preserved). The robust morphotype (from both BEB500 and TCH1065) differs from J. curtedulensis in the absence of clear phalangeal pads (preservation bias?), the absence of the peculiar, isolated proximal pad PIII1 of digit III, and the interdigital divarication angles (asymmetric vs. symmetric); it also differs from the Morphotype II tracks in the absence of blunt digit impressions, possible evidence of a discrete phalangeal pad, and the presence of clear claw marks. Therefore, despite some overlap of the morphometric data and the fact that there are considerable differences in shape (although some of them might be extramorphological variations), we consider that there are not enough data to support the second hypothesis just on the basis of the footprint shape and morphometric data.

Finally, we examine whether there is any spatio-temporal relationship between the larger and the smaller tracks from the Ajoie ichnocoenosis. Lockley (1994) warned that the track data “that most probably represent monospecific assemblages are those obtained for a single ichnotaxon from a single bedding plane.” In this regard, it is interesting to note the scarcity of large theropod tracks in the ichnoassemblages where both the gracile and the robust morphotype have been identified, mainly levels BEB500, TCH1065, and CRO500. Level BEB500 (Fig. S1), the one with the highest number of studied tracks (n = 39), is mainly composed of sauropods (n = 17 trackways), and minute to small tridactyl (n = 158 trackways) tracks. No tracks assigned to J. curtedulensis or M. transjuranicus have been documented in this level although it is the surface with the most Morphotype II tracks (n = 8 trackways) documented. Level TCH1065 (Fig. S2) (n = 15 studied tracks) is composed of 189 tracks, mainly of minute to small-sized theropods, and two parallel trackways (TCH1065-T26, TCH1065-T27) assigned to Jurabrontes have also been documented. In level CRO500 (Fig. S3) (n = 20 studied tracks), 16 sauropod trackways, and 57 tridactyl trackways have been documented. One of the tridactyl trackways (CRO500-T43) has been assigned to Morphotype II (Razzolini et al., 2017). Thus, there are in three cases a large track type (Morphotype II in BEB500 and CRO500, and Jurabrontes in TCH1065) and the robust and the gracile morphotypes in the same surface (Figs. S1–S3). Interestingly, no Megalosauripus tracks have been documented in any of the three levels. One way to confirm that some of the small tracks were juveniles of the larger ichnospecies would be to find some kind of relationship among them, such as gregarious behaviour (sensu Castanera et al., 2014). However, there is no clear relationship between the trackways, neither of the same size nor of different sizes. In BEB500 (Fig. S1), trackways TR1, TR3, TR4, TR5, TR6, and TR8 (Morphotype II) cross several trackways made by small trackmakers, but the orientations are completely different and do not show any kind of relationship. TR2 (Morphotype II) is subparallel with T34 (small track but unknown morphotype) at the beginning of the trackway but shows a significant change in direction, so this does not show any relationship either. Notably, TR7 (Morphotype II) is a long trackway that is subparallel to T120 (robust morphotype). Tracks T120-L10 and T120-R10 tread over tracks TR7-R8 and TR7-L9 but pass afterwards, so although this might indicate some kind of interaction there is no clear evidence of gregarious behavior. In level TCH1065 (Fig. S2), the two parallel trackways (TCH1065-T26, TCH1065-T27) assigned to Jurabrontes do not show any evidence of a relationship with the smaller tracks either. Finally, in CRO500 (Fig. S3), T43 (Morphotype II) is slightly subparallel to T42 (small track but unknown morphotype), but there is no clear evidence to suggest that they were walking together. To sum up, generally the orientation of the large trackways does not seem to suggest any sort of relationship, with the possible exception of TR7 and T120. This single case might hint at the hypothesis that some tracks of the robust morphotype (BEB500-T120) might represent a juvenile of the producer of the tracks classified as Morphotype II. However, BEB500-T120 is the very trackway that shows more morphometric similarities to Jurabrontes than to Morphotype II (Fig. 6), thus weakening this hypothesis. In the light of the previous discussion, there is no evidence to suggest an interaction (i.e., behavioral aspect) among the dinosaurs that produced trackways of different sizes when they are left on the same surface. Thus, there is no indication (nor in footprint shape, nor morphometric, nor spatio-temporal) to suggest that the gracile and the robust morphotype are smaller tracks of the same of the larger morphotypes described from the area. Thus, the differences between the larger and the smaller morphotypes have thus led us to treat them as different ichnotaxa (see next section).

Ichnotaxonomy

As noted by Marty (2008), small to medium-sized tridactyl tracks are generally not very common in the Late Jurassic and Early Cretaceous, and accordingly such tracks have only recently been the focus of ichnotaxonomic descriptions. Although recent descriptions have considerably increased the number of small to medium-sized tridactyl tracks, few ichnotaxa have been described in the Late Jurassic and Early Cretaceous of Europe. Lockley, Meyer & Moratalla (2000) suggested that theropod track morphologies are much more variable through time than previously thought. These authors pointed out that “the perception of morphological conservatism and uniformity through time is, in part, a function of lack of study of adequately large samples of well-preserved material (Baird, 1957).” In this sense, the studied tracks from the Ajoie ichnocoenosis represent a good sample of tridactyl dinosaur tracks in terms of the number of specimens (n = 93), with a considerable quality of preservation in many of them (n = 23 with a preservation grade greater than 2).

Although they are not very abundant in other European tracksites, small to medium-sized tridactyl trackways are the most abundant in the Ajoie ichnocoenosis. As mentioned above, the main small to medium-sized tridactyl dinosaur ichnotaxa that have been described from the Late Jurassic of Europe are (Fig. 7) Grallator (Fig. 7A) and Anomoepus (Fig. 7B) in Spain (Lockley et al., 2008; Piñuela, 2015; Castanera, Piñuela & García-Ramos, 2016); Carmelopodus (Fig. 7C) and Eubrontes (Fig. 7D) in France (Mazin et al., 2000; Mazin, Hantzpergue & Pouech, 2016); Wildeichnus (Fig. 7E), cf. Jialingpus (Fig. 7F) and Dineichnus (Fig. 7G) in Poland (Gierliński, Niedźwiedzki & Nowacki, 2009); Dineichnus (Fig. 7H) (Lockley et al., 1998a) and Therangospodus-like tracks (Fig. 7I) (Lockley, Meyer & Moratalla, 2000) in Portugal; and Grallator in Germany (Fig. 7J) (Diedrich, 2011). In addition, Conti et al. (2005) described medium-sized footprints (Fig. 7K) that “resemble Therangospodus” (their type 3) and another morphotype (their type 2, Fig. 7L) that shares the same functional character with Carmelopodus, i.e., the lack of the fourth proximal pad on digit IV.

Figure 7 Main small-medium-sized tridactyl dinosaur footprints described in the Late Jurassic of Europe.

(A) Grallator from Spain (S, after Castanera, Piñuela & García-Ramos, 2016); (B) Anomoepus from Spain (S, after Piñuela, 2015); (C) Carmelopodus from France (C, after Mazin, Hantzpergue & Pouech, 2016); (D) Eubrontes from France (C, after Mazin et al., 2000); (E) Wildeichnus from Poland (C, after Gierliński, Niedźwiedzki & Nowacki, 2009); (F) cf. Jialingpus from Poland (C, after Gierliński, Niedźwiedzki & Nowacki, 2009). (G) Dineichnus from Poland (C, after Gierliński, Niedźwiedzki & Nowacki, 2009); (H) Dineichnus from Portugal (S, Lockley et al., 1998a); (I) Therangospodus-like track from Portugal (S, after Lockley, Meyer & Moratalla, 2000); (J) Grallator from Germany (S, after Diedrich, 2011); (K) Therangospodus-like track from Italy (C, after Conti et al., 2005); (L) Carmelopodus-like track from Italy (C, after Conti et al., 2005). Scale bar = 1 cm (E), 5 cm (A, F, G), 10 cm (B, C, D, H, I, J, K, L). S and C refer to siliciclastic and carbonate substrate, respectively.

When compared with the type specimens of these ichnotaxa, the new data on the gracile morphotype of CRO500-T10 (Figs. 5A and 8N) (see previous sections) allow us to rule out the presence of Carmelopodus untermannorum (Fig. 8A) in the Ajoie, as previously discussed. Generally, the gracile morphotype (Figs. 8M–8O) does not fit with key features of the diagnosis of this ichnotaxon (Lockley et al., 1998b), differing in the phalangeal pad formula (2-3-4 rather than 2-3-3), symmetry, different length/width ratio, or the lower divarication. Among other theropod ichnotaxa, the gracile morphotype shows considerable differences with respect to Wildeichnus navesi (Fig. 8B, Casamiquela, 1964; Valais, 2011) from the Jurassic of Argentina (as well as larger size, a not subequal but lower divarication angle, larger claw marks, an unrounded digital phalangeal pad in digit IV, greater asymmetry, a generally higher length/width ratio); and with respect to Therangospodus pandemicus from the Late Jurassic of North America and Asia (Fig. 8C, smaller size, presence of clear phalangeal pads, higher mesaxony) (Lockley, Meyer & Moratalla, 2000; Fanti et al., 2013). The differences with respect to ornithopod ichnotaxa are noteworthy: it differs from Anomoepus scambus (Fig. 8D) in being less symmetric, having a metatarsal-phalangeal pad of digit IV not in line with the digit III axis, no hallux marks, higher mesaxony, and no manus prints present (see Olsen & Rainforth, 2003). It also differs notably with respect to Dineichnus socialis (Fig. 8E) for higher FL/FW ratio, higher mesaxony, no quadripartite morphology, a different heel pad impression, lower digit divarication (see Lockley et al., 1998a).

Figure 8 Small-medium-sized tridactyl dinosaur ichnotaxa with affinities with the described morphotypes.

(A) Outline drawing of the holotype of Carmelopodus untermannorum (S, redrawn after Lockley et al., 1998b); (B) outline drawing of the holotype of Wildeichnus navesi (V, redrawn after Lockley, Mitchell & Odier, 2007); (C) outline drawing of the topotype of Therangospodus pandemicus (S, after Lockley, Meyer & Moratalla, 2000); (D) outline drawing of Anomoepus scambus (S, after Olsen & Rainforth, 2003); (E) outline drawing of the holotype of Dineichnus socialis (S, after Lockley et al., 1998a); (F) composite outline drawing of type trackway of Grallator parallelus (S, redrawn from Olsen, Smith & McDonald, 1998); (G) outline drawing of type specimen of Anchisauripus sillimani (S, redrawn from Olsen, Smith & McDonald, 1998); (H) outline drawing of type specimen of Eubrontes giganteus (S, redrawn from Olsen, Smith & McDonald, 1998); (I) outline drawing of type specimen of Jialingpus yuechiensis (S, redrawn from Lockley et al., 2013); (J) outline drawing of type specimen of Kalohipus bretunensis (S, redrawn from Fuentes Vidarte & Meijide Calvo, 1998); (K) drawing of type specimen of Jurabrontes curtedulensis (redrawn from Marty et al., 2017); (L) outline drawing of type specimen of Megalosauripus transjuranicus (redrawn from Razzolini et al., 2017); (M) outline drawing of specimen BSY1020-E2 (cf. Kalohipus isp.); (N) outline drawing of specimen CRO500-T10-L10 (cf. Kalohipus isp.); (O) outline drawing of specimen TCH-1060-E58 (cf. Kalohipus isp.); (P) outline drawing of specimen TCH-1065-T21-R1 (cf. Therangospodus isp.); (Q) outline drawing of specimen BEB500-T120-R5 (Therangospodus? isp.). S, C, and V refer to siliciclastic, carbonate and volcanoclastic substrate, respectively. Scale bar = 2 cm (B, D), 5 cm (F, G, H, I, J), 10 cm (A, C, E, L, M–Q), 50 cm (K).

The features of the gracile morphotype fit better with the smaller ichnotaxa of the Grallator–Anchisauripus–Eubrontes (Figs. 8F–8H) plexus (Olsen, 1980; Demathieu, 1990; Weems, 1992; Olsen, Smith & McDonald, 1998): small to medium-sized, well-defined digital pads, digits II and IV of similar length, digit III being longer and showing high mesaxony, an oval/subrounded “heel” and a low interdigital angle. Although these footprints have mainly been described from Late Triassic and Early-Middle Jurassic deposits, in recent years they are also known from younger strata including the Late Jurassic of Europe (see Castanera, Piñuela & García-Ramos, 2016 and references therein). Regarding the use of the ichnotaxon Anchisauripus, Castanera, Piñuela & García-Ramos (2016) wrote a short review examining how different authors have considered Grallator and Anchisauripus as synonyms (Lucas et al., 2006; Lockley, 2009; Piñuela, 2015). The main sample of “grallatorid” tracks that has been described from Late Jurassic deposits in Europe comes from Asturias (Spain), and these have been assigned to Grallator (Castanera, Piñuela & García-Ramos, 2016). However, the gracile tracks from the Ajoie ichnocoenosis differ in the digit proportions (FL/FW ratio) and mesaxony (Fig. 9) from those in Asturias. It should be noted that the Asturian sample shows a great variation in mesaxony (that does not correlate with size). This holds also true for the gracile morphotype although the footprint proportions are less variable. Whereas Castanera, Piñuela & García-Ramos (2016) stated that mesaxony “should be used with caution in distinguishing between different ichnotaxa,” we consider that the differences in mesaxony between the gracile morphotype (AT = 0.53–0.98) and the Grallator tracks from Asturias (AT = 0.72–1.12, Castanera, Piñuela & García-Ramos, 2016) tracks are large enough to do so. Furthermore, the FL/FW ratio is also considerably higher in the Grallator tracks (FL/FW ratio = 1.73–2.5, Castanera, Piñuela & García-Ramos, 2016) than in the gracile morphotype (FL/FW ratio = 1.50–1.90). Oversplitting has occurred in some theropod ichnotaxa similar to Grallator–Eubrontes plexus. For example, Lockley et al. (2013) proposed a great reduction in the Jurassic theropod ichnotaxa from Asia, arguing that many of them were subjective junior synonyms of Grallator and Eubrontes. Nonetheless, the authors retain the ichnotaxon Jialingpus yuechiensis (Fig. 8I) from the Late Jurassic-Early Cretaceous of China (Xing et al., 2014). On the basis of digit proportions (FL/FW ratio) and mesaxony, the gracile morphotype falls partially within the range of Jialingpus but also within the range of K. bretunensis (Fig. 8J) from the Early Cretaceous (Berriasian) of Spain (Fuentes Vidarte & Meijide Calvo, 1998; Castanera et al., 2015). According to Xing et al. (2014), the main differences between Jialingpus and Grallator are the presence of a digit I trace and the large metatarsophalangeal area positioned in line with digit III, which are its main features. These features are absent in the gracile morphotype, so an assignment to Jialingpus can be excluded. On the other hand, the diagnosis of K. bretunensis (Fuentes Vidarte & Meijide Calvo, 1998) clearly includes features that distinguish it from the gracile morphotype, such as its smaller size or robust digits, and as seen in Fig. 9, the footprint proportions and especially the mesaxony are also slightly different. As mentioned above, the morphology is also different from the larger ichnotaxa (J. curtedulensis, Fig. 8K, and M. transjuranicus, Fig. 8L) that occur in the same deposits.

Figure 9 Bivariate graph plotting the footprint length/footprint width ratio against Mesaxony of the studied tracks (gracile and robust morphotype) with some of the main dinosaur tridactyl ichnotaxa mentioned in the text.

Outline drawings not to scale.

To summarize, the gracile morphotype is quite similar to other grallatorid tracks (Grallator, Anchisauripus, Kalohipus, Jialingpus), the main differences being the digit proportions and mesaxony. Given the current state of knowledge, it is difficult to interpret how much variation between the aforementioned ichnotaxa is a consequence of variations in preservation, ontogeny or ichnodiversity. Taking into account the whole discussion, and bearing in mind the high variation in both the FL/FW ratio and mesaxony seen in tracks assigned to Grallator, we thus tentatively classify the gracile morphotype as cf. Kalohipus isp., as this is the ichnotaxon that is closest to it (Fig. 9). Future studies should elucidate the similarities and differences between these grallatorid tracks. Jialingpus tracks have been also described in the Late Jurassic/Early Cretaceous of Europe (Gierliński, Niedźwiedzki & Nowacki, 2009), and an analysis of the differences between Jialingpus and other grallatorid tracks (including Kalohipus) is “pending” (Xing et al., 2014). In this regard, it would be interesting to note the differences in mesaxony among both Kalohipus and Jialingpus (low mesaxony) and Grallator (high mesaxony), and questioning whether mesaxony is a good measure for discriminating between the three ichnotaxa. Possible substrate related differences in preservation or ichnofacies substrates have to be tested too. For example, K. bretunensis and the main grallatorid ichnotaxa (Figs. 8F–8J) are preserved in siliciclastic sediments whereas the tracks cf. Kalohipus isp. from the Swiss Jura Mountains are preserved in marginal marine carbonates.

Regarding the robust morphotype (Figs. 8P–8Q), a crucial question is whether it represents a single ichnotaxon. In this context, it should be noted that, the morphology of the tracks with a preservation grade of 2 or more, as well as the footprint proportions (Fig. 6B) such as those of trackway BEB500-T120 and the tracks from TCH1065 (TCH1065-T21-R1, TCH1065-E124, and TCH1065-E188) varies considerably. The appearance of this morphotype is completely different from ichnotaxa like C. untermannorum (Fig. 8A, size, phalangeal pad formula, digit divarication, well-developed claw marks), W. navesi (Fig. 8B, size, gracility, symmetry, length/width ratio, and mesaxony), A. scambus (Fig. 8D, size, absence of a manus impression, morphology of the metatarsal-phalangeal pad of digit IV) and D. socialis (Fig. 8E, no quadripartite morphology or circular heel pad impression). It differs also from all the aforementioned grallatorid ichnotaxa Grallator–Anchisauripus–Eubrontes, plus Jialingpus, Kalohipus (Figs. 8F–8J, mainly in the more robust morphology, footprint proportions, mesaxony, heel morphology, divarication), and the larger ichnotaxa (J. curtedulensis, Fig. 8K, and M. transjuranicus, Fig. 8L) that occur at the same localities.

Of all the known ichnotaxa, it shares most similarities with T. pandemicus (Fig. 8C, Lockley, Meyer & Moratalla, 2000; Fanti et al., 2013; see also Castanera et al., 2013b for new data on tracks previously assigned to Therangospodus), although the robust morphotype from the Swiss Jura Mountains has a higher digit divarication and probably higher mesaxony (unpublished data for this parameter in the original publication, Lockley, Meyer & Moratalla, 2000). According to the original diagnosis, this ichnotaxon is a “medium sized, elongate, asymmetric theropod track with coalesced, elongate, oval digital pads, not separated into discrete phalangeal pads. Trackway narrow with little or no rotation of digit III long axis from trackway axis”. The tracks from the Ajoie ichnoassemblages are slightly smaller in size than T. pandemicus (Lockley, Meyer & Moratalla, 2000; Fanti et al., 2013). According to these authors, and based on the original descriptions by Lockley, Meyer & Moratalla (2000), Therangospodus is characterized by: “(1) oval digital pads not separated into discrete digital pads, (2) no rotation of digit III, (3) narrow trackway, and (4) relatively reduced size (<30 cm in average length).” Regarding the absence of discrete digital pads, Lockley, Meyer & Moratalla (2000) described in the type ichnospecies of Therangospodus the presence of “faint indentations at the margin of the pads” that sometimes reveal the location of the phalangeal pads, suggesting a 2-3-4 phalangeal pad formula. Razzolini et al. (2017) also pointed out the difficulties of distinguishing between Therangospodus and Megalosauripus. This has also been previously discussed by other authors (Gierliński, Niedźwiedzki & Pieńkowski, 2001; Piñuela, 2015), suggesting that some of the diagnostic features might be extramorphological variations. It is notable that Megalosauripus and Therangospodus generally co-occur in the same sites (Meyer & Lockley, 1997; Lockley, Meyer & Moratalla, 2000; Lockley, Meyer & Santos, 2000; Xing, Harris & Gierliński, 2011; Fanti et al., 2013), which might be relevant as the size and preservation could be the only differences between the two ichnotaxa. Interestingly, as we have seen in the previous section, the robust morphotype does not co-occur with any Megalosauripus tracks, although some of them (BEB500-T120) co-occur with tracks described as Morphotype II. Even though the robust morphotype is reminiscent of T. pandemicus, it is not possible to assign it to this ichnospecies or to any of the known small-medium-sized ichnotaxa. The rarity of collected specimens and the preservation grade (none of them as high as 2.5–3) prevents us from erecting a new ichnotaxon. Taking into account that T. pandemicus is the closest ichnotaxon described, we thus tentatively classify the tracks from level TCH1065 as cf. Therangospodus isp. and the tracks from BEB500 as Therangospodus? isp. in order to show that there are some differences (both morphometric and in shape) within the robust morphotype. As for the Swiss specimens, T. pandemicus tracks have been preserved in carbonate materials (Lockley, Meyer & Moratalla, 2000), so we can rule out the differences between this ichnotaxon and the robust morphotype being a consequence of this factor as the substrate and the palaeonvironmental conditions were probably similar at the time of track production.

Conclusion

The minute to medium-sized tridactyl dinosaur footprints from the tracksites of Highway A16 in the Jura Mountains (NW Switzerland) represent one of the largest samples from the Late Jurassic worldwide. The integrated analyses of the quality of preservation (preservation grade), the maximum depth, the shape variation along the trackway, and the footprint proportions (FL/FW ratio and mesaxony) open a new window into the interpretation of dinosaur track variations also considering other factors such as preservation, ontogeny and ichnotaxonomy. The descriptions and analyses of the material have made it possible to characterize in detail two different morphotypes, one gracile and one robust, that were already identified in the field. The new data allow us to rule out the notion that the two morphotypes represent a morphological continuum of extramorphological variations, or ontogenetic variations of the larger tracks described from the same sites. New morphometric data allow us to include the small sized tracks previously described from Chevenez–Combe Ronde tracksite within the gracile morphotype, being the only case in the studied sample that might be explained by an ontogenetic variation. An ichnotaxonomical comparison with the main minute to medium-sized tridactyl ichnotaxa did not allow assigning them to any known ichnotaxon with confidence. The gracile morphotype, though similar to some grallatorid ichnotaxa, shows a number of morphometric differences and have been assigned to cf. Kalohipus isp. The robust morphotype, though similar to T. pandemicus, also shows some differences with respect to the diagnosis of the type specimen, and therefore is classified as cf. Therangospodus isp. and Therangospodus? isp. Further work is needed in order to understand the possible influence of the substrate composition on theropod ichnotaxonomy in general and the aforementioned ichnotaxa in particular. This study also highlights the difficulties of distinguishing between minute and medium-sized tridactyl dinosaur ichnotaxa and the importance of analyzing different factors related to preservation and ontogeny before assigning a single track to a specific ichnotaxon. The new data increase theropod ichnodiversity to 4/5? theropod ichnotaxa in the tidal flats of the Jura carbonate platform and support previous suggestions that carbonate tidal flats were mainly dominated by theropod and sauropod dinosaurs (Lockley, Hunt & Meyer, 1994; D’Orazi Porchetti et al., 2016).

Supplemental Information

Supplemental Information 1 List of the specimens analysed, their quality of preservation (preservation grade) and the maximum depth.

Those with preservation grade 0–0.5 are not included in the figshare file. The tracks where the variation along the trackway has been analysed are in red.

Click here for additional data file.

Supplemental Information 2 Map of the Courtedoux—Béchat Bovais tracksite, level 500 (BEB500).

In red (gracile) and blue (robust) the minute to medium-sized tridactyl tracks and in green, the larger morphtoype (Morphotype II). Source credit: OCC-SAP, Canton Jura.

Click here for additional data file.

Supplemental Information 3 Map of the Courtedoux—Tchâfouè tracksite, level 1065 (TCH1065).

In red (gracile) and blue (robust) the minute to medium-sized tridactyl tracks and in green, the larger morphtoype (Jurabrontes curtedulensis see Marty et al., 2017). Source credit: OCC-SAP, Canton Jura.

Click here for additional data file.

Supplemental Information 4 Map of the Chevenez—Combe Ronde, level 500 (CRO500).

In red (gracile) and blue (robust) the minute to medium-sized tridactyl tracks and in green, the larger morphtoype (Morphotype II). Source credit: OCC-SAP, Canton Jura.

Click here for additional data file.

We thank all technicians, photographers, geometers, drawers, collection managers, and preparators of the PALA16 that were involved during the excavation and documentation of the tracksites and during the set-up and organization of the track collection. We also thank the scientific staff of the PALA16 and JURASSICA Muséum for various stimulating discussions and valuable input. The authors also thank Laura Piñuela, Vanda Santos, Ignacio Díaz-Martínez, Oliver W. M. Rauhut, and Novella L. Razzolini for fruitful discussion on the topic of this manuscript. The comments of the editor (Claudia Marsicano) and the reviewers (Jesper Milàn and Verónica Krapovickas) have considerably improved the quality of the manuscript and are greatly appreciated. Rupert Glasgow revised the English grammar.

Additional Information and Declarations

Competing Interests

Author Contributions

Data Availability

The authors declare that they have no competing interests.

Diego Castanera conceived and designed the experiments, performed the experiments, analyzed the data, contributed reagents/materials/analysis tools, prepared figures and/or tables, authored or reviewed drafts of the paper, approved the final draft.

Matteo Belvedere conceived and designed the experiments, performed the experiments, analyzed the data, contributed reagents/materials/analysis tools, prepared figures and/or tables, authored or reviewed drafts of the paper, approved the final draft.

Daniel Marty conceived and designed the experiments, performed the experiments, analyzed the data, contributed reagents/materials/analysis tools, prepared figures and/or tables, authored or reviewed drafts of the paper, approved the final draft.

Géraldine Paratte performed the experiments, contributed reagents/materials/analysis tools, prepared figures and/or tables, approved the final draft.

Marielle Lapaire-Cattin performed the experiments, contributed reagents/materials/analysis tools, prepared figures and/or tables, approved the final draft.

Christel Lovis performed the experiments, contributed reagents/materials/analysis tools, prepared figures and/or tables, approved the final draft.

Christian A. Meyer conceived and designed the experiments, performed the experiments, analyzed the data, contributed reagents/materials/analysis tools, prepared figures and/or tables, authored or reviewed drafts of the paper, approved the final draft.

The following information was supplied regarding data availability:

Belvedere, Matteo; Castanera, Diego; Marty, Daniel; Paratte, Géraldine; Cattin, Marielle; Lovis, Christel; Meyer, Christian A. (2018): A walk in the maze: Variation in Late Jurassic tridactyl dinosaur tracks from the Swiss Jura Mountains (NW Switzerland). 3D photogrammetric models. figshare. https://doi.org/10.6084/m9.figshare.5662306.v2.

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
