# Peer review of "A walk in the maze: variation in Late Jurassic tridactyl dinosaur tracks from the Swiss Jura Mountains (NW Switzerland)"

_PeerJ, doi:10.7717/peerj.4579_

## Round 0.1 · original submission · Major Revisions

Dear Dr. Castanera

Your Ms # 22250 entitle "A walk in the maze: Variation in Late Jurassic tridactyl dinosaur tracks - A case study from the Late Jurassic of the Swiss Jura Mountains (NW Switzerland)" co-authored with Belvedere, Marty, Paratte, Lapaire-Cattin, Lovis & Meyer have been reviewed by two reviewers and myself as Editor. I consider that your Ms needs Major Revisions before to be consider for publication in PeerJ.

My main concern is about the Ms discussion as some sections need to be re-written for clarification as the text is convoluted. It would benefit from sharpening and tightening the text. Particularly, the section "Morphotype variation due to ontogeny" needs to be better sustained and the rationale behind your hypothesis more clearly define. Please see carefully the annotated PDF attached with my particular comments on this matter.

I also recommend you to check the use of the language in the Ms by a native English speaker before re-submission.

Accordingly, I am requesting that you revise your Ms taking particular attention to the points mentioned above.

Thank you for submitting your Ms to PeerJ and I look forward to receiving your revision.

Sincerely,

Claudia Marsicano

·

Basic reporting

This is a very well written and well documented study. My only concern is that it is simply too long and exhausting to read, but as I have a hard time pointing out specific sections that could be omitted, without weaking the argumentetion I will recommedn it for publication in its present form, as PeerJ has no page limitations.
The figures are well executed and are all nessecery to support the results presented in the paper.

I have not cross checked the refernce list.

Experimental design

no comment

Validity of the findings

no comments

Additional comments

This is a very well-executed piece of research and it clearly demonstrates how usefull a tool 3D photogrammetry has become in palaeoichnological imterpretations.

·

Basic reporting

The English grammar used through the manuscript is clear and professional. Although, some sentences and expressions could be improved in order to sum up the most important statements. I made a few comments on the pdf version. However, as I am not a native English speaker my grammar corrections are limited. Please consider a final check by a native speaker or a grammar expert.

The introduction and background are well presented and completed with actualized literature.

The figures are relevant, high quality, and well labeled. Although, I found figure 1B to small so it is difficult to read it, and a label in red color for no apparent reason on figure 6. Both comments are also included on the pdf version.

The raw data is supplied.

Experimental design

The research is original, with primary data analysis. The research questions are clearly stated, relevant, and meaningful and followed throughout the manuscript. The technical methods used to analyze the data are modern, proper and well described to replicate it.
There are some comments on the approach of the morphotype variation due to ontogeny. See below.

Validity of the findings

The manuscript is a good example of how to analyze in detail the ichnological record of a unit plentiful of tracks and trackways with similar morphologies preserved. The authors evaluated the shape variations of footprints along trackways and between morphotypes, the possible ontogenetic relationship between them, evaluate its ichnotaxonomy to finally contribute to the faunistic composition of the ichnoassemblage.
The most important comment is on the discussion about morphology variations due to ontogeny (lines 397-492). The authors explore to lines of thought 1) variations in morphology, 2) spatiotemporal relationship. The second line of thought is perfectly well stated. I find some problems with the first one.
On the first paragraph (lines 398-423) the author mentioned different criteria used by other authors to evaluate ontogenetic variations. One seems to involve morphologic variations on footprints due to allometric growth of individual as the Grallator-Eubrontes plexus suggested by Olsen, 1980; Olsen, Smith & McDonald, 1998; Moreau et al., 2012(lines 401-410). The other involves linear growth (lines 420-423). On the following paragraphs the author analyzed the small tracks from the Swiss Jura Mountains. Those paragraphs are confusing sense they do not express clearly which kind of growth they expect to find (isometric or allometric). Besides, results somehow difficult to read because it abounds in descriptions and comparisons but lacks of sentences that clearly summarize the conclusions arrived. See as example the comment about the paragraph (424-443) on the pdf version. In that sense, it would help to state more clearly the lines of thought and to reinforce the discussion with sentences that sum up the most important statements. I consider these are important modifications due to the topic is of high interest and has been only superficially aboard on the ichnological literature.
Other minor comments are expressed on the pdf version.
Finally, the conclusions are well stated and linked to the original research questions.

Additional comments

no comment

---

## Round 0.2 · accepted · Accept

Dear Dr. Castanera

I am pleased to inform you that your manuscript # 22250 entitle "A walk in the maze: Variation in Late Jurassic tridactyl dinosaur tracks - A case study from the Late Jurassic of the Swiss Jura Mountains (NW Switzerland)" co-authored with Belvedere, Marty, Paratte, Lapaire-Cattin, Lovis & Meyer is now accepted for publication in PeerJ.

Thank you again for considering PeerJ and we look forward to your future contributions to the Journal.

sincerely,

Claudia Marsicano